# Heterogeneous Impact of Social Integration on the Health of Rural-to-Urban Migrants in China

**DOI:** 10.3390/ijerph19169999

**Published:** 2022-08-13

**Authors:** Haiyang Lu, Ivan T. Kandilov, Peng Nie

**Affiliations:** 1Institute of Western China Economic Research, Southwestern University of Finance and Economics, Chengdu 610074, China; 2Department of Agricultural and Resource Economics, North Carolina State University, Raleigh, NC 27695, USA; 3School of Economics and Finance, Xi’an Jiaotong University, Xi’an 710061, China; 4Institute for Health Care & Public Management, University of Hohenheim, 70599 Stuttgart, Germany

**Keywords:** social integration, health, dialectal diversity, internal migration, China, heterogeneous effects

## Abstract

Background: While several studies have found that lower levels of social integration may lead to a deterioration in the health status of migrants, previous research on the nexus between social integration and health has generally ignored the potential endogeneity of social integration. This paper examines the heterogeneous impact of social integration on the health of rural-to-urban migrants in China by exploiting plausibly exogenous, long-term, geographic variation in dialectal diversity. Methods: Drawing on nationally representative data from the 2017 China Migrants Dynamic Survey (n = 117,446), we first regressed self-reported health on social integration using ordinary least squares estimation and then used an ordered probit model as a robustness check. Additionally, to rule out the potential endogeneity of social integration, we relied mainly on an instrumental variable approach and used dialectal diversity as a source of exogenous variation for social integration. Results: We found that social integration has a significant positive impact on rural-to-urban migrants’ health. We also detected considerable heterogeneity in the effects of social integration across gender, generation, and wage levels: the health status of women, more recent generation migrants, and migrants with wages in the middle of wage distribution are more likely to be affected by social integration. Conclusions: We confirmed the beneficial impact of social integration on migrants’ health, which has some important policy implications. Successful migration policies should take the fundamental issue of migrants’ social integration into account.

## 1. Introduction

Since the Opening of China in 1978, the country has witnessed a rapid urbanization process, with a tremendous number of rural workers migrating from the poor countryside to affluent urban areas [1,2]. In particular, as documented by Su et al. [3], the number of urban populations in China has increased dramatically by more than 500 million since 1979, and more than 75% of this growth is due to rural-to-urban migration. 

There is little doubt that the hukou system has had a profound influence on China’ internal migration. The Chinese hukou system, established in 1955, is generally considered as the basis of China’s dualistic socio-economic structure and rural–urban dual citizenship [4]. This system registers household members according to the hukou type (rural or urban) and the place of household registration. Specifically, all Chinese households are divided into rural (agricultural) and urban (nonagricultural) households, and individuals are categorized as either insiders (individuals with local hukou) or outsiders (individuals without local hukou) [5]. It is essentially a migration control system that functions very much like an international border, identifying aliens and excluding them from pension benefits, legal aids, medical services and access to public schools [6]. Under the hukou system, the migration behavior and integration status of rural-to-urban migrants are found to be affected by factors such as human capital [7], labor market outcomes [8], and social exclusion [5].

The massive internal migration in China has elicited intense interest in the health concerns of rural-to-urban migrants in host societies, with particular attention to the determinants of migrants’ health [9,10,11]. A growing literature has documented that migration per se [12], environmental hazards [13], social networks [14,15], social stigma and discrimination [16], and utilization of health services [17] are significant predictors of the physical and mental health of rural-to-urban migrants. 

One strand of the literature most closely related to our study looks specifically at the health consequences of social integration [18,19]. Social integration is recognized as a multidimensional concept, and there is no clear and unified definition among researchers in different fields. Durkheim [20] defined social integration as the means by which individuals connect, interact, and validate each other within a community. Other seminal work, including Gordon [21], Alba and Nee [22], and Berry [23], regarded social integration as a process of assimilation or acculturation. Moreover, social integration also refers to an individual’s attachment to a society, commonly considered as a mechanism for sustaining social norms and reducing deviant behaviors through formal links to social institutions as well as informal ties to friends and families [24,25]. 

Several sets of classical theories, such as Bowlby’s attachment theory [26] and the acculturation theory developed by social psychologists [27], have laid the foundation for the empirical studies on social integration and its impact on health. Undoubtedly, the pioneering work of Durkheim has made an immeasurable contribution to the literature on the link between society and health, in particular, on how social integration and cohesion affect mortality [28]. Empirically, using a sample of China’s internal migrants, Lin et al. [6] found that components of social integration, such as integration willingness, economic integration, and acculturation were positively correlated with migrants’ health. Similarly, Brydsten et al. [29] documented that social integration was associated with mental health inequality between the migrants and natives in Sweden. 

Although existing studies find a strong correlation between social integration and health-related outcomes based on different cultural contexts, few studies take into account the endogeneity of social integration. Our work enriches the existing literature in the nexus between social integration and migrants’ health in two respects: First, we identify the impact of social integration on migrants’ health by exploiting the exogenous long-term, geographic variation in dialectal diversity in China. Specifically, we regress the health of rural-to-urban migrants on social integration and instrument the latter by two indicators of dialectal diversity: the number of sub-dialects and a dialectal fractionalization index at the city level. We show that both metrics of dialectal diversity have significant explanatory power over social integration, whilst the two instruments have no direct influence on rural-to-urban migrants’ health.

Second, we contribute to the literature by providing a nationally representative analysis of the impact of social integration on health for Chinese internal migrants. China offers an interesting and unique case for investigating how social integration influences the health of rural-to-urban migrants, mainly because it is recognized as a country with the largest internal migration in the world [30]. Moreover, the unique household registration system (the hukou system) hinders the process of the social integration of China’s internal migrants [31]. Despite being Chinese citizens, rural-to-urban migrants are generally excluded from the welfare and civic entitlements provided by the state due to the lack of local hukou status [32]. Thus, it is worthwhile to investigate the health consequences of social integration in the Chinese context.

The next section reviews the relevant literature. Section 3 describes the data and the methods we employ in empirical analysis. In Section 4, we present our empirical results. Section 5 discuss our key findings and Section 6 concludes.

## 2. Literature and Hypotheses

Durkheim’s theory implies that social integration plays a crucial role in ensuring that individuals are socially connected, regardless of interpersonal relationships, and that they are highly attached to commonly held beliefs and values [33]. Correspondingly, the weakening of a society’s capacity for integration caused by rapid social change or political turmoil may cause alienation, anomie, and, in particular, the occurrence of suicide [20]. The seminal work of Durkheim and other theorists has led to extensive discussions about the major ramifications of social integration, ranging from violence and homicide [34,35,36] to physical and mental health [37,38]. 

There is substantial research in sociology and health economics that exploits, but does not yield a firm conclusion on, the link between social integration and health-related metrics. Most empirical analyses have produced positive results [18,39]. The mechanisms behind the positive relationship between social integration and health consequences include social support, access to resources (e.g., health services, job opportunities), and social influence (e.g., social control, norms) [28,40]. However, the work by Steel et al. [38] points out that social integration is not always beneficial for the health of individuals. A deeper exploration of the literature on social networks offers potential interpretations for the deleterious effect of social integration on health. As highlighted by the work of Smith and Christakis [41] and Kushner and Sterk [42], it is misleading to believe that social networks are intrinsically good because they may also have negative impacts. In particular, health-risk behaviors (e.g., smoking, alcohol, and drug use) shared through social networks may lead to health deterioration [43,44,45]. For example, the work by Young and Rice [46] documents that the use of social networking sites is likely to be associated with increased sexual risk behaviors. This finding is consistent with Powell et al. [47], who suggest that some behaviors related to obesity are socially transmissible. 

It is worth noting that the papers in this literature, including those we discussed above, differ in their measurement of social integration and choice of specifications, and most studies do not tackle the issue of endogeneity of social integration. One exception is the relevant work by Appau et al. [48], who find that integrating into a community is likely to increase the subjective well-being (SWB) of UK citizens. They use ethnic diversity as an instrument for social integration because they argue that a high level of ethnic diversity is related to low levels of social capital, which leads to low integration. Ethnic diversity, however, can plausibly exert a direct impact on SWB. For example, Akay et al. [49] argue that ethnic diversity can affect SWB by enhancing the productivity of natives. If, indeed, ethnic diversity exerts a direct impact, it would be problematic to use it as an instrument for social integration.

Due to the structural and institutional restrictions imposed by the Hukou system and other local protective policies, rural-to-urban migrants in China typically encounter substantial difficulties in their efforts to integrate into the host society [50]. They are commonly referred to as marginalized groups in urban societies, with limited employment opportunities, poor housing conditions, and insufficient public health services [51]. Regarding health, Hu et al. [52] document that Chinese rural-to-urban migrants are mainly subject to health concerns related to maternal health, occupational health, and infectious diseases. 

As Rose et al. [19] highlighted, individuals are social beings whose lives are inevitably affected by the nature of their social relations. The stronger the attachment to a social group, the more likely individuals are to follow the norms of the group, not only as guidelines for their behaviors, but also as an emotional bond to maintain important social ties [53]. Conversely, weak attachment among group members may lead to low levels of supportive relationships, resulting in worse psychosocial and socio-economic consequences [54]. According to Berkman et al. [28], the structural arrangements of social systems determine the resources available to individuals and thus determine their emotional and behavioral responses. Because social integration may change the health behviors of rural-to-urban migrants and enable them to access local health services and social support, we propose the following hypothesis and research question about the nexus between social integration and health among rural-to-urban migrants in China:

**H1.** 
*Social integration has a positive impact on the health of rural-to-urban migrants in China.*


**R1.** 
*How does the impact of social integration vary by gender, generation, and wage status?*


## 3. Materials and Methods

### 3.1. Data

The data for our analysis come from the 2017 wave of the China Migrants Dynamic Survey (CMDS), conducted by the National Health and Family Planning Commission of China. The CMDS is a nationally, representative, cross-sectional survey of internal migrants over the age of 15 who had lived at the surveyed sites (cities) for at least one month prior to the survey. A stratified, multistage sampling design with probability proportional to size was used in the survey. 

A total of 169,989 migrants from 351 cities in all 31 mainland provinces and municipalities completed the survey. In our analysis, we focus on rural-to-urban migrants with rural hukou and 16–65 years of age. Observations with missing values on core metrics (including sub-indexes of social integration and health) are dropped from the sample. As a result, 117,446 observations were used for analysis after sample selection and data cleaning. The sample size of the regression analyses varies according to the different specifications of the model, depending on the data availability of the variables involved. 

### 3.2. Measures of Health

Self-reported health (SRH), an often-used proxy in the health literature [10], is the main health metric in this paper. SRH can effectively capture an individual’s overall health and is also regarded as a valid metric for morbidity, fitness, and loss of functional ability [55]. Respondents in CMDS (2017) were asked how they evaluated their health status on a 4-point scale ranging from “excellent” to “very poor”. We coded the responses so that higher values represent better health. As a robustness check, we further adopted several dichotomous measures of health, alternatively, from the series of questions: “Have you had any morbidity symptoms in the last year? (i) fever, (ii) diarrhea, (iii) heavy cold”.

### 3.3. Social Integration

Social integration includes various dimensions such as sense of belonging, integration willingness, social interaction, acculturation, and self-identity [56]. In this paper, eight questions were used to capture these dimensions (see Table A1). On a 4-point scale from “1 = totally disagree” to “4 = totally agree”, the respondents were asked to rate how much they agree that: (i) “I like this city”; (ii) “I noticed the changes in this city”; (iii) “I am willing to integrate into this city”; (iv) “The local residents are willing to accept me”; (v) “I have not been discriminated against by local citizens”; (vi) “Following the customs of my hometown is not important to me”; (vii) “My personal hygiene habits are not different from those of local citizens”; and (viii) “I am already a local resident”. A composite score for social integration is computed, summing up all responses to the eight questions, and total score ranges between 8 to 32, with a higher score indicating a higher level of social integration. The Cronbach’s alpha coefficient for the eight sub-indexes is 0.734, indicating a sufficient reliability of our composite indicator of social integration.

### 3.4. Other Covariates and Descriptive Statistics

Given that many other individual characteristics that are correlated with health might also be associated with social integration, following prior work [24,57], we control for several other covariates, including gender, age, education, marriage status, length of stay in the city, household migration status, the logarithm of the monthly wage, weekly work hours, medical insurance status, and social ties. In addition, we also include dummy variables for occupation and city, as well as logged GDP per capita at the city level. The data on GDP per capita come from the China City Statistical Yearbook. The descriptive statistics of the variables are presented in Table A2.

### 3.5. Empirical Strategies

To explore the nexus between social integration and rural-to-urban migrants’ health, we estimate the following equation:(1)$$HS_{i}=\beta SI_{i}+\gamma X_{i}+u_{i}$$
where HS is the proxy for health status, SI is the indicator of social integration, X is a vector of controls, and u is the error term. Although our core metric of health, SRH, is ordinal in nature, many prior studies treat it as a linear variable [58] and estimate the regression model using the standard OLS method. In a methodological paper, Ferrer-i-Carbonell and Frijters [59] highlight that treating the measure of the outcome variable as cardinal or ordinal generally does not lead to different findings. To facilitate analysis, we first regress SRH on social integration using OLS and then we also use an ordered probit (OP) model as a robustness check.

To rule out the potential endogeneity of social integration, we rely mainly on an instrumental variables (IV) approach and use dialectal diversity, measured by the number of sub-dialects (Div_n) and a dialectal fractionalization index (Div_m) as a source of exogenous variation for social integration. Our data on dialectal diversity come from Liu et al. [60]. They first collected data on the number of sub-dialects in each prefecture-level city in China based on the *Dictionary of Chinese Dialect* and the *Language Atlas of China*. Further, they constructed a Herfindahl-type fractionalization index [61] by weighting the population according to the following formula: $Div\text{\_}m_{j}=1-\sum_{n=1}^{Div\text{\_}n}S_{nj}^{2}$, where $S_{nj}$ is the share of the population who use the sub-dialect n in city j. 

The dialectal fractionalization index captures the probability that two randomly selected individuals use different sub-dialects, with a larger $Div\text{\_}m_{j}$ indicating a more diverse local dialect. We observe that there are large differences in the dialectal fractionalization index of cities with the same number of sub-dialects. For example, whilst both Beijing and Tongren have three sub-dialects, the dialectal fractionalization index of the two cities is 0.256 and 0.772, respectively.

The intuition behind the IV strategy in our context is that dialectal diversity may hinder social integration due to greater linguistic barriers [62] and potential political cleavages [63], while it is unlikely to exert a direct influence on individual health. Based on data from Australia, Leigh [64] reports that ethno-linguistic fractionalization is negatively related to localized trust. The work by Fieldhouse and Cutts [65], however, suggests that linguistic diversity may also have positive effects because individuals living in a society with high diversity may be more tolerant and less hostile to outsiders. This argument is, to some extent, supported by Tolsma et al. [66], who demonstrate that wealthier and better-educated people living in diverse neighborhoods tend to be more tolerant and have more contact with their neighbors. 

Notably, dialectal diversity may affect health through other macro-level channels, in particular the income distribution. Many economists and social scientists, including Sturm and De Haan [67] and Houle [68] document the correlation between linguistic fractionalization and income redistribution. To control for this, we include logged GDP per capita at the city level as a regressor. 

Naturally, it is impossible to control for all possible channels through which social integration affects the health of migrants. Following Acemoglu et al. [69], we employ two strategies to substantiate that the number of sub-dialects and the dialectal fractionalization index serve as valid instruments. First, we include the two instruments directly as control variables, and examine the impact of dialectal diversity on health. Second, we check the validity of our instruments by using overidentification tests. In addition, as a robustness check, we use IV-ordered probit (IV-OP) and propensity score matching (PSM) approaches to test the reliability of our findings.

## 4. Results

### 4.1. Effects of Social Integration on Health

Table 1 presents the results from the OLS regressions. We start by regressing SRH on social integration without conditioning on other covariates (column 1). Column 2 reports the estimated effect of social integration on SRH, conditioning on the observables listed in Table A2. In columns 3 and 4 of Table 1, we further control for occupational and city fixed effects. Overall, we find that gender, education, logged monthly wage, social ties, and logged GDP per capital are positively associated with SRH, while age, length of stay in this city, household migration, and work hour are negatively associated with SRH. In particular, the results show that conditioning on the various observables leaves the positive and significant effect of social integration on SRH of rural-to-urban migrants essentially unchanged. For example, column 4 shows that one standard deviation increase in social integration is related to an increase of 0.072 standard deviations in SRH.

To address the potential endogeneity issue, we next use the number of sub-dialects and the dialectal fractionalization index as instrumental variables for social integration. Before employing the IV approach, we regress SRH on the two instruments to check if dialectal diversity has a direct impact on rural-to-urban migrants’ SRH. The results are encouraging. In column 1 of Table 2, we find that, conditional on other observables, neither the number of sub-dialects nor the dialectal fractionalization index exerts a significant impact on SRH. The IV estimates are then presented in columns 2 and 3. Both the overidentification test and first-stage F-test confirm that our IVs are valid. As shown in column 3, one standard deviation in social integration increases the SRH score by 0.304 standard deviations, which is substantially larger in magnitude than the OLS estimates. These findings imply that the OLS estimates presented in Table 1 tend to be biased downwards due to endogeneity. Thus, H1 is supported.

### 4.2. Robustness Checks

Our first robustness check concerns the alternative measures of health status. In Table 3, we report the 2SLS estimates of the effect of social integration on the health of rural-to-urban migrants using a variety of proxies for morbidity, other than SRH, as the dependent variable. The results demonstrate an adverse impact of social integration on the four metrics of morbidity, implying that the beneficial effect of social integration on health are robust to different health measures.

In Table 4, we further performed several tests to check if our results are robust to different specifications. We first re-ran an IV estimation, controlling for additional covariates that may be correlated with the health among rural-to-urban migrants, including social security card ownership status, knowledge of the National Basic Public Health Services Project (NBPHSP), and attendance of health education. Our main results remained unchanged after adjusting for those additional controls (Panel A). In Panels B and C, we re-estimate the effect of social integration on SRH using OP and IV-OP approaches, respectively. The results are very similar to those reported in Table 2. In Panels D and E of Table 4, we produce a new composite index for social integration by exploratory factor analysis and construct a dichotomous measure of SRH (1 = excellent, 0 = otherwise), respectively. The results are still very similar to our main results. In Panel F, we examine whether our results are sensitive to different sample restrictions by excluding those who have college degree. In Panel G, we further exclude those who have weak settlement intention. Once again, the positive effect of social integration remains significant. Given that sample selection bias may also lead to biased estimates [70], we finally use a PSM method to estimate the average treatment effects on the treated (ATT). As shown in Panel H of Table 4, the estimated ATT is positive and statistically significant. 

### 4.3. Heterogeneous Effects

The effect of social integration on SRH may vary across different subpopulations. In response to RQ1, in Table 5, we explore whether the impact of social integration on SRH differs by gender, by generation, and by income. Columns 1 and 2 of Table 5 show a significant positive impact of social integration on SRH for female migrants, but no impact for male migrants. In China, rural-to-urban migrants who were born after 1980 are generally classified as younger generation or new-generation migrants [8]. We present the results for new-generation and older-generation migrants in Columns 3 and 4 of Table 5, respectively, and find that the impact of social integration is significant and stronger only for new-generation migrants, whereas we find no significant impact for older-generation migrants. Furthermore, we partition the sample into three wage groups by the tertile of the monthly wage distribution, and we find that the significant positive effect of social integration is mainly pronounced for the migrants in the middle wage tertile (columns 5, 6, and 7 of Table 5).

## 5. Discussion

Although many believe that social integration plays an important role in explaining differences in the health of migrants, the few empirical studies on the health–social integration gradients are far from conclusive, especially for developing countries with large internal migration. In this paper, we examine the heterogenous impact of social integration on the health of rural-to-urban migrants in China. Our work is among the first to rule out the potential endogeneity of social integration by exploiting plausibly exogenous variation in dialectal diversity.

Using the number of sub-dialects and a dialectal fractionalization index at the city level as instrumental variables for social integration, we found that social integration has a significant, positive impact on the health of rural-to-urban migrants in China. This finding is supported by Wen et al. [71], who showed that integration into the neighborhood was closely related to the health of internal migrants in China. The most likely explanation is that social integration can change the health consciousness or behavior of migrants [72]. Due to the difficulty of establishing new social relations and integrating into urban society, rural-to-urban migrants in China were found to easily resort to smoking and drinking to relieve loneliness and provide life pressure [73]. As highlighted by Berkman et al. [28], increasing social disconnection is often related to the cumulative prevalence of health-threatening behaviors such as alcohol and tobacco consumption, physical inactivity, and obesity. As such, the beneficial health effect of social integration may be partly due to changes in the health behaviors of rural-to-urban migrants. It is also likely that the positive impact of social integration on the health of rural-to-urban migrants is largely a result of increased utilization of health services. As noted by Liang et al. [74], internal migrants who are more socially integrated are more likely to establish health records and have health education.

We also revealed that the effect is heterogeneous across different subpopulations. In particular, the positive impact of social integration is more pronounced for female migrants and new-generation migrants. One possible explanation for the gender difference is that women have higher emotional needs and social network involvement than men. As documented by Kawachi and Berkman [75], compared with men, women are more likely to establish and maintain emotionally close relationships, which in turn may make them more susceptible to the “contagion of stress” when stressful life events affect those with whom they are emotionally close. In terms of the generational difference, new-generation migrants are generally better educated, have higher expectations for life in the host society, and are less tolerant of harsh working conditions than the older-generation migrants (those who were born before 1980) [8]. Yang [5] points out that, in addition to making money, the new-generation migrants also have a strong desire to live like urbanites and to integrate into urban societies. Thus, it is reasonable that the impact of social integration is greater for new-generation migrants. These results echo the work by Lin et al. [6], who suggested that female and young migrants may need more attention with regard to social integration. 

It is also interesting to note that migrants in the middle of the wage distribution are more likely to be affected by social integration in terms of health. One possible explanation is that, compared with low-wage migrants (people who generally have low expectations for social integration due to lack of certain skills) and those with high wages (individuals who can easily achieve social integration without too much effort), middle-wage migrants may have higher expectations for social integration because their aspirations are more commensurate with their abilities. As such, they are affected more by social integration in terms of health outcomes.

Several limitations of our work warrant discussion. First, although we believe that the impact of social integration estimated from IV regressions can technically be interpreted as causal, this study is intrinsically a cross-sectional analysis and therefore fails to identify the long-term impact of social integration on rural–urban migrants. Second, all health measures were self-reported, which means that the measurement error of these indicators is greater than objective health indicators (such as blood test reports). In fact, the measurement errors of subjective proxies (e.g., subjective well-being and generalized trust) are generally unavoidable, because respondents usually need to answer survey questions in a short time, and some participants may conceal their true ideas. Third, due to data unavailability of mental health, we mainly focused on the impact of social integration on physical health. Migration and social integration issues may affect mental health more than physical health. Thus, future research on assessing the effect of social integration on both mental and physical health is needed. Moreover, we were unable to explicitly examine the mechanisms through which social integration may operates on health due to the absence of data. Future studies will benefit from large-scale longitudinal data that contain detailed information on objective health measures and allow for examining the effects of social integration over a longer time period. Additionally, probing into the mediating role of some channel variables (such as health behaviors, access to material resources, and social engagement) may enrich our understanding of the relationship between social integration and the health of migrants.

## 6. Conclusions

Prior studies regarding the health of China’s rural-to-urban migrants have mainly focused on the effect of migration per se, while we emphasize the aspect of social integration, which has been largely neglected by existing research. Our findings confirmed the beneficial impact of social integration on internal migrants’ health in China, and these findings have some important policy implications. Successful migration policies should take the fundamental issue of migrants’ social integration into account. It is necessary to comply with the Copenhagen Declaration on Social Development, which emphasizes the promotion of social integration by eliminating all forms of discrimination and protecting disadvantaged and vulnerable groups and individuals [76]. In practice, it may be beneficial for governments and communities to provide migrants with proper interventions, such as language courses, non-academic seminars, or community activities, so as to foster interaction between natives and migrants, which can help improve migrants’ health. More importantly, since the hukou system has long been recognized as the fundamental cause preventing China’s rural-to-urban migrants from integrating into urban society, it is necessary to further reduce the barriers and the costs of hukou conversion for rural-to-urban migrants. Institutional reforms that aim to increase migrants’ access to urban benefits, such as housing and health services, can also help reduce health inequality. This issue is particularly urgent in China, where the population is aging at an unprecedented pace and the demographic dividend is diminishing [77].

## Figures and Tables

**Table 1 ijerph-19-09999-t001:** OLS estimates for the impact of social integration on SRH.

	DV: SRH			
	(1)	(2)	(3)	(4)
Social integration	0.006 *** (0.000)	0.007 *** (0.000)	0.007 *** (0.000)	0.009 *** (0.000)
	[0.046]	[0.057]	[0.058]	[0.072]
Gender		0.028 *** (0.003)	0.028 *** (0.003)	0.029 *** (0.003)
Age		−0.008 *** (0.000)	−0.007 *** (0.000)	−0.008 *** (0.000)
Junior middle school		0.055 *** (0.005)	0.052 *** (0.005)	0.050 *** (0.005)
Senior middle school		0.050 *** (0.005)	0.047 *** (0.005)	0.047 *** (0.005)
College		0.035 *** (0.006)	0.034 *** (0.006)	0.035 *** (0.006)
Married		0.004 (0.003)	0.004 (0.004)	0.015 *** (0.004)
Length of stay in this city		−0.003 *** (0.000)	−0.003 *** (0.000)	−0.003 *** (0.000)
Household migration		−0.008 *** (0.003)	−0.007 *** (0.003)	−0.006 *** (0.003)
Logged monthly wage		0.038 *** (0.003)	0.034 *** (0.003)	0.032 *** (0.003)
Work hour		−0.000 *** (0.000)	−0.001 *** (0.000)	−0.001 *** (0.000)
Medical insurance		−0.001 (0.003)	−0.002 (0.003)	0.001 (0.003)
Social ties		0.006 ** (0.003)	0.006 ** (0.003)	0.013 *** (0.003)
Logged GDP per capita		0.012 *** (0.003)	0.013 *** (0.003)	0.055 * (0.003)
Constant	3.654 *** (0.010)	3.448 *** (0.037)	3.473 *** (0.042)	3.133 *** (0.372)
Occupational dummies	No	No	Yes	Yes
City dummies	No	No	No	Yes
R-squared	0.002	0.056	0.058	0.092
Observations	117,446	99,139	99,139	99,139

Note: * *p* < 0.1, ** *p* < 0.05, *** *p* value < 0.01. Standard errors are shown in parentheses and standardized coefficients are in brackets.

**Table 2 ijerph-19-09999-t002:** Tests for the validity of the two instruments and the IV estimates.

	(1)	(2)	(3)
	OLS	First Stage: Social Integration	Second Stage: SRH
Dialectal fractionalization index	−0.003	−0.252 ***	
	(0.009)	(0.071)	
Number of sub-dialects	0.004	0.125 ***	
	(0.002)	(0.018)	
Social integration	0.007 ***		0.038 **
	(0.000)		(0.018)
			[0.304]
Constant	3.395 ***	29.989 ***	2.482 ***
	(0.049)	(0.325)	(0.555)
Controls	Yes	Yes	Yes
Overidentification test		0.404	
First stage F-test		24.755 ***	
Durbin-Wu-Hausman chi-square test		2.889 *	
Observations	89,806	89,806	89,806

Note: * *p* < 0.1, ** *p* < 0.05, *** *p* < 0.01. Standard errors are shown in parentheses and standardized coefficients are in brackets.

**Table 3 ijerph-19-09999-t003:** Robustness checks: alternative measures of health.

	Morbidity Last Year	Fever	Diarrhea	Heavy Cold
Social integration	−0.130 ***	−0.070 ***	−0.029 ***	−0.142 ***
	(0.022)	(0.013)	(0.012)	(0.023)
	[−0.851]	[−0.716]	[−0.283]	[−0.938]
Constant	3.179 ***	2.340 ***	1.106 ***	5.021 ***
	(0.685)	(0.414)	(0.375)	(0.717)
Controls	Yes	Yes	Yes	Yes
Observations	89,806	89,806	89,806	89,806

Note: * *p* < 0.1, ** *p* < 0.05, *** *p* < 0.01. Standard errors are shown in parentheses and standardized coefficients are in brackets.

**Table 4 ijerph-19-09999-t004:** Robustness checks: alternative specifications.

Coefficient	ATT	**Standard Error**	Observations
** *Panel A: Include additional control variables* **			
0.032 **		0.017	77,010
** *Panel B: OP estimates* **			
0.030 ***		0.002	99,139
** *Panel C: IV-OP estimates* **			
0.023 ***		0.005	99,139
** *Panel D: Generate a new composite measure for social integration by an exploratory factor analysis* **			
0.178 **		0.085	89,806
** *Panel E: Dichotomous measure of SRH, IV-probit* **			
0.213 **		0.041	89,806
** *Panel F: Only include rural-to-urban migrants with education lower than college* **			
0.037 *		0.022	78,676
** *Panel G: Only include rural-to-urban migrants with education lower than college and have strong settlement intention* **			
0.040 *		0.022	76,711
** *Panel H: Dichotomous measure of social integration and use the PSM to estimate the ATT* **			
	0.028 ***	0.004	99,141

Note: OP = ordered probit, IV-OP = instrumental variable ordered probit, SRH = self-reported health, IV-probit = instrumental variable probit, PSM = propensity score matching, ATT = average treatment effects on the treated. * *p* < 0.1, ** *p* < 0.05, *** *p* < 0.01.

**Table 5 ijerph-19-09999-t005:** Heterogeneous effects.

	Gender		Generation		Wage Tertile		
	Male (1)	Female (2)	New-Generation (3)	Older-Generation (4)	1st (5)	2nd (6)	3rd (7)
Social integration	0.001	0.088 ***	0.119 ***	−0.048	0.017	0.071 *	0.060
	(0.025)	(0.033)	(0.030)	(0.035)	(0.027)	(0.038)	(0.040)
	[0.008]	[0.696]	[1.198]	[−0.320]	[0.127]	[0.585]	[0.529]
Constant	3.518 ***	0.915	0.160	4.949 ***	3.011 ***	1.414	2.037 *
	(0.743)	(1.010)	(0.917)	(1.011)	(0.866)	(1.197)	(1.214)
Controls	Yes	Yes	Yes	Yes	Yes	Yes	Yes
Observations	51,355	38,451	53,594	36,212	38,628	20,165	31,013

Note: * *p* < 0.1, ** *p* < 0.05, *** *p* < 0.01. Standard errors are shown in parentheses and standardized coefficients are in brackets.

## Data Availability

The datasets used and analyzed during the current study are available at https://www.chinaldrk.org.cn, accessed on 30 January 2022.

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
