# Peer review of "Heterogeneous Impact of Social Integration on the Health of Rural-to-Urban Migrants in China"

_ijerph, 2022, doi:10.3390/ijerph19169999_

Round 1
Reviewer 1 Report
Although I find this work very promising, I would suggest to Authors to:
- shorten the abstract
- integrate the 2.1 in the Introduction
- move the tables at the bottom of the paper, at least Descriptive statistics and tests.
Author Response
We are grateful to the editors and the anonymous referees for many insightful suggestions, which helped improve our paper. We have tried as best as we can to respond to each of them directly.
Reviewer #1
Although I find this work very promising, I would suggest to Authors to:
- shorten the abstract
Response: Thank you for this comment. Following your comment, we have shortened the abstract.
- integrate the 2.1 in the Introduction
Response: Thank you for this comment. Adopting your comment, we have now moved the 2.1 to the Introduction (see lines 32-100).
- move the tables at the bottom of the paper, at least Descriptive statistics and tests.
Response: Thanks for pointing this out. Following your suggestion, we have moved the original Tables 1 and 2 to the Appendix (see lines 440-447).
Reviewer 2 Report
The paper is well -written and the research design adheres to the theoretical discussion. A very solid paper which will surely contributes to the study of rural-urban migration in China.
What the writers can do is that they should discuss their results referring to the existing literature they have used. However, it seems that when they are doing the discussion, they have not been referring to the literature they have been using and comparing and contrasting the results. Discussion should be rewritten in a way that it reflects there is a connection between the theoretical discussions and the findings.
Author Response
We are grateful to the editors and the anonymous referees for many insightful suggestions, which helped improve our paper. We have tried as best as we can to respond to each of them directly.
The paper is well -written and the research design adheres to the theoretical discussion. A very solid paper which will surely contributes to the study of rural-urban migration in China.
Response: Thanks very much for this positive comment.
What the writers can do is that they should discuss their results referring to the existing literature they have used. However, it seems that when they are doing the discussion, they have not been referring to the literature they have been using and comparing and contrasting the results. Discussion should be rewritten in a way that it reflects there is a connection between the theoretical discussions and the findings.
Response: Thank you for the advice. We have polished up the “Discussion” section to compare our findings with the literature and also find more connections between our findings and other earlier works (lines 335-402).
Reviewer 3 Report
Please find the referee report attached.

Author Response
We are grateful to the editors and the anonymous referees for many insightful suggestions, which helped improve our paper. We have tried as best as we can to respond to each of them directly.
Reviewer #3
Using the 2017 China Migrants Dynamic Survey (CMDS) data, this paper offers credible evidence to claim that improved social integration causes better health. Some other recently published articles, not cited here, provide further evidence to support similar claims. However, the novel instrument of social integration that this paper constructed from two indicators of dialectal diversity and its IV regression-based evidence make the findings more credible than the papers with similar results published earlier. The paper is very well written. It makes its case convincingly. However, I have added a few comments for improvement.
Response: Thank you for your insightful comments and suggestions on our paper. We have made substantial changes to the paper based on your suggestions. Below we take this opportunity to address each of your specific concerns.
Some Comments and Suggestions:
- Discussions through lines 250 – 258 address conflicting theories of how diversity may affect social integration. In particular, the debate suggests that it is plausible that the impact varies significantly between the following two groups: (i) educated and wealthy and (ii) uneducated and poor. Naturally, a question arises about how the findings reported in Tables 4 – 7 may vary between these two groups. By examining the data of international migrants in China, Fan, Yan, and Yan (2020) reported that a formal educational experience in China helped improve "the self-rated health status." The paper will make a significantly more substantial contribution if it can shed light
on the marginal impact of social integration on health after controlling for education and wealth.
Response: Thank you for this comment. As you suggested, we controlled for education and the logged monthly wage (as a measure of wealth) in all models. The descriptive statistics of the variables are presented in Table A2 (lines 443-447) and our OLS and IV estimates (see Tables 1 and 2) facilitated us to interpret the estimated coefficients as the effect size of social integration on health after adjusting for education, wealth and other controls.
- Liang et al. (2020) emphasized the role of public institutions in facilitating social integration through structural integration. They use the same dataset (2017 CMDS) as the one used in this paper. Naturally, some comparisons with their findings may enhance the paper's marginal contribution. If possible, it would be valuable for us to learn if social integration matters for health even after controlling for public institutions.
Response: Thank you for this comment. In the original manuscript, the impact of public institutions can be captured by medical insurance status and city-level fixed effects. In this revision, as a robustness check, we controlled additional covariates of public institutions that may be correlated with the health among rural-to-urban migrants, including social security card ownership status, knowledge of National Basic Public Health Services Project (NBPHSP) and attendance of health education. (lines 299-304)
- The gender aspect is intriguing. The paper could elaborate on whether the gender difference remains significant even after controlling for education, wealth, and public institutions.
Response: Thank you for this comment. As you suggested, we have already controlled for education, wealth (proxied by the logged monthly wage), and public institutions (proxied by medical insurance status and city dummies) in our analysis of heterogeneous impacts. Columns 1 and 2 of Table 5 show a significant positive impact of social integration on SRH for female migrants, but no impact for male migrants. And we have provided detailed explanations for this gender difference in the Discussion part:
“One possible explanation for the gender difference is that women have higher emotional needs and social network involvement than men. As documented by Kawachi & Berkman [75], compared with men, women are more likely to establish and maintain emotionally close relationships, which in turn may make them more susceptible to the “contagion of stress” when stressful life events affect those with whom they are emotionally close.” (lines 361-366)
- The argument that "the hukou system" deters social integration appears ad hoc before presenting evidence that migrants overcome that language-dialect barrier through education, wealth, and public institution. So, the abstract need not push that argument.
Response: Thank you for pointing this out. Following your suggestion, we have removed this statement in the abstract when revising the paper.
References
- Liang, J.; Shi, Y.; Osman, M.; Shrestha, B.; Wang, P. The Association between Social Integration and Utilization of Essential Public Health Services among Internal Migrants in China: A Multilevel Logistic Analysis. Int. J. Environ. Res. Public Health 2020, 17, 6524. https://doi.org/10.3390/ijerph17186524
- Fan, X.; Yan, F.; Yan, W. Better Choice, Better Health? Social Integration and Health Inequality among International Migrants in Hangzhou, China. Int. J. Environ. Res. Public Health 2020, 17, 4787. https://doi.org/10.3390/ijerph17134787
- Kawachi, I.; Berkman, L. F. Social ties and mental health. J. Urban Health 2001, 78, 458-467. https://doi.org/10.1093/jurban/78.3.458
Reviewer 4 Report
This reviewer commends the authors for their study titled, “Heterogeneous Impact of Social Integration on the Health of Rural-to-Urban Migrants in China.”
Introduction
There needs to be a description of the hukou system in China included in the Introduction Section. It would also be helpful if the Introduction Section included a narrative on the determinants of rural-urban migration and social integration in China.
Method and data Analysis
What is/are the research hypothesis/ Research question (s).How was sample selection conducted? What are the inclusion and exclusion criteria ?
What is the validity and reliability of the Self-Reported Health survey instruments?
What is the validity and reliability of the Social Integration Questionnaire/survey instrument?
What is the outcome/dependent variable and the independent variables in this study? Migration and social integration issues may affect mental health more than physical health. It is not clear if the central focus of this study is health status of migrants or healthcare access to migrants. This needs to be clearly stated.
How much variance in the outcome is explained by each of the independent variables?
Results
There needs to be a narrative summary of the results that includes all the significant findings for each of the tables, and the data depicted therein.
Conclusion
Apart from some exercise with large data, the study does not provide new information or new knowledge that could influence social policy in China. China is a member of the UN and signatory to the Copenhagen Declaration on Social Development and already committed to matters of social integration. Please read the reference provided below, especially page 10 of the article about commitment to social integration.
Reference
World Summit for Social Development. (1995). Copenhagen declaration on social development. Available at https://www.un.org/en/development/desa/population/migration/generalassembly/docs/globalcompact/A_CONF.166_9_Declaration.pdf
Author Response
We are grateful to the editors and the anonymous referees for many insightful suggestions, which helped improve our paper. We have tried as best as we can to respond to each of them directly.
Reviewer #4
This reviewer commends the authors for their study titled, “Heterogeneous Impact of Social Integration on the Health of Rural-to-Urban Migrants in China.”
Introduction
There needs to be a description of the hukou system in China included in the Introduction Section. It would also be helpful if the Introduction Section included a narrative on the determinants of rural-urban migration and social integration in China.
Response: Thank you for this comment. We have now done this based on your suggestion (lines 38-50).
Method and data Analysis
What is/are the research hypothesis/ Research question (s). How was sample selection conducted? What are the inclusion and exclusion criteria?
Response: Thank you for your valuable comments. we have formulated formal hypothesis/research question based on your comments (lines 157-160). In our analysis, we focus on rural-to-urban migrants with rural hukou and those aged 16-65 years old. Observations with missing values on core metrics (including sub-indexes of social integration and health) are dropped from the sample. As a result, 117,446 observations were used for analysis after sample selection and data cleaning. The sample size of the regression analyses varies according to the different specifications of the model, depending on data availability of the variables involved (lines 169-175).
What is the validity and reliability of the Self-Reported Health survey instruments?
Response: Health status is considered a multi-dimensional concept and difficult to measure. For this reason, various proxies are used to conceptualise and measure health. Self-reported health (SRH) is one of the most commonly used indicators in the health literature (e.g., Lu et al., 2020; Pan et al., 2016). It can reflect an individual’s overall health status comprehensively. It is also recognised as a valid proxy for fitness, loss of functional ability, and morbidity (Idler & Kasl, 1995). In addition to use SRH as the main health measure in this paper, as a robustness check, we further adopt several dichotomous measures of health, based on a series of questions: “Whether you had any morbidity symptoms in the last year? (i) fever, (ii) diarrhea, (iii) heavy cold.”
In the “Discussion” section, we acknowledge this as one limitation of this research:
“……all health measures were self-reported, which means that the measurement error of these indicators is greater than objective health indicators (such as blood test reports). In fact, the measurement errors of subjective proxies (e.g., subjective well-being and generalized trust) are generally unavoidable, because respondents usually need to answer survey questions in a short time, and some participants may conceal their true ideas.” (lines 391-395)
What is the validity and reliability of the Social Integration Questionnaire/survey instrument?
Response: Thank you for this comment. When updating the paper, we have clarified the validity and reliability of the Social Integration Questionnaire/survey instrument: “The Cronbach’s alpha coefficient for the eight sub-indexes is 0.734, indicating a sufficient reliability of our composite indicator of social integration.” (lines 197-198)
What is the outcome/dependent variable and the independent variables in this study? Migration and social integration issues may affect mental health more than physical health. It is not clear if the central focus of this study is health status of migrants or healthcare access to migrants. This needs to be clearly stated.
Response: Thank you for your comments. We fully agree that social integration may have a larger effect on mental health compared to physical health. Since the information on mental health is unavailable, this study focused on the physical health among rural migrants in this study (see lines 177-184). Of course, we also acknowledged this as an additional limitation in the Discussion part:
“Third, due to data unavailability of mental health, we mainly focused on the impact of social integration on physical health. Migration and social integration issues may affect mental health more than physical health. Thus, future research on assessing the effect of social integration on both mental and physical health is needed.” (lines 391-395)
How much variance in the outcome is explained by each of the independent variables?
Response: Based on your suggestion, we have incorporated R-Squared into the OLS estimation, which helped us to assess the variance in the outcome explained by each of the controls (see Table 1).
Results
There needs to be a narrative summary of the results that includes all the significant findings for each of the tables, and the data depicted therein.
Response: Thank you for this good point. In this revision, we have done this for Table 1, including the estimates for all the coefficients:
“Overall, we find that gender, education, logged monthly wage, social ties and logged GDP per capital are positively associated with SRH, while age, length of stay in this city, household migration, and work hour are negatively associated with SRH. In particular, the results show that conditioning on the various observables leaves the positive and significant effect of social integration on SRH of rural-to-urban migrants essentially un-changed.” (lines 262-268)
Conclusion
Apart from some exercise with large data, the study does not provide new information or new knowledge that could influence social policy in China. China is a member of the UN and signatory to the Copenhagen Declaration on Social Development and already committed to matters of social integration. Please read the reference provided below, especially page 10 of the article about commitment to social integration.
Reference
World Summit for Social Development. (1995). Copenhagen declaration on social development. Available at https://www.un.org/en/development/desa/population/migration/generalassembly/docs/globalcompact/A_CONF.166_9_Declaration.pdf
Response: Thank you for this great comment. Although our IV approach goes a further step to address endogeneity issue of social integration, we acknowledge that this paper has several limitations, one of which is the lack of policy innovation. We agree that the Copenhagen Declaration on Social Development included several useful guidelines on social integration, but mostly at the macro level. In this revision, we have cited the reference recommended by the reviewer to provide detailed suggestions for the social integration of migrant workers in China in the Discussion part:
“It is necessary to comply with the Copenhagen Declaration on Social Development, which emphasizes the promotion of social integration by eliminating all forms of discrimination and protecting disadvantaged and vulnerable groups and individuals [76]. In practice, it may be beneficial for governments and communities to provide migrants with proper interventions, such as language courses, non-academic seminars or community activities, so as to foster interaction between natives and migrants, which can help improve migrants’ health. More importantly, since the hukou system has long been recognized as the fundamental cause preventing China’s rural-to-urban migrants from integrating into urban society, it is necessary to further reduce the barriers and the costs of hukou conversion for rural-to-urban migrants. Institutional reforms that aim to increase migrants’ access to urban benefits, such as housing and health services, can also help reduce health inequality. This issue is particularly urgent in China, where the population is aging at an unprecedented pace and the demographic dividend is diminishing.” (lines 409-422)
References:
Idler, E. L., & Kasl, S. V. (1995). Self-ratings of health: do they also predict change in functional ability? The Journals of Gerontology Series B: Psychological Sciences and Social Sciences, 50(6), S344-S353.
Lu, H., Kandilov, I. T., & Zhu, R. (2020). The impact of internal migration on the health of rural migrants: Evidence from longitudinal data in China. The Journal of Development Studies, 56(4), 840-855.
Pan, J., Lei, X., & Liu, G. G. (2016). Health insurance and health status: exploring the causal effect from a policy intervention. Health Economics, 25(11), 1389-1402.
World Summit for Social Development. Copenhagen declaration on social development. 1995. Available at https://www.un.org/en/development/desa/population/migration/generalassembly/docs/globalcompact/A_CONF.166_9_Declaration.pdf
Round 2
Reviewer 4 Report
The authors have accepted and responded appropriately to the concerns and comments provided by this reviewer.